  

# SMA-linked SMN mutants prevent phase separation properties and SMN interactions with FMRP family members

Olivier Binda[1,2] ⓘ, Franceline Juillard[1], Julia Novion Ducassou[3], Constance Kleijwegt[1,4] ⓘ, Geneviève Paris[2], Andréanne Didillon[2], Faouzi Baklouti[1] ⓘ, Armelle Corpet[1] ⓘ, Yohann Couté[3] ⓘ, Jocelyn Côté[2], Patrick Lomonte[1] ⓘ

**Although recent advances in gene therapy provide hope for spinal muscular atrophy (SMA) patients, the pathology remains the leading genetic cause of infant mortality. SMA is a monogenic pathology that originates from the loss of the *SMN1* gene in most cases or mutations in rare cases. Interestingly, several *SMN1* mutations occur within the TUDOR methylarginine reader domain of SMN. We hypothesized that in *SMN1* mutant cases, SMA may emerge from aberrant protein-protein interactions between SMN and key neuronal factors. Using a BioID proteomic approach, we have identified and validated a number of SMN-interacting proteins, including fragile X mental retardation protein (FMRP) family members (FMR_FM). Importantly, SMA-linked SMN_TUDOR mutant forms (SMN_ST) failed to interact with FMR_FM. In agreement with the recent work, we define biochemically that SMN forms droplets in vitro and these droplets are stabilized by RNA, suggesting that SMN could be involved in the formation of membraneless organelles, such as Cajal nuclear bodies. Finally, we found that SMN and FMRP co-fractionate with polysomes, in an RNA-dependent manner, suggesting a potential role in localized translation in motor neurons.**

## Introduction

Although the loss of the survival motor neuron 1 gene (*SMN1*) was identified in 1995 to be responsible for spinal muscular atrophy (SMA) (Lefebvre et al, 1995), SMA still remains the leading genetic cause of infant mortality. Importantly, whereas gene therapy expands health-span and life-span, SMA remains without a cure (Chaytow et al, 2021). SMN, as a protein, is translated in humans from two genes, the telomeric *SMN1* and a centromeric duplication *SMN2*. Both *SMN1* and *SMN2* encode the exact same functional SMN protein. However, the *SMN2* duplication contains, among other changes, a pyrimidine transition (cytosine to thymine) in exon 7 that introduces an exonic splicing silencer element, leading to the prevailing exclusion of exon 7 (SMN_Δ7) and subsequent production of a truncated, unstable, and rapidly degraded protein (Lorson & Androphy, 2000).

SMN is characterized by a notorious domain called TUDOR. The TUDOR domain is part of a large, still expanding family of histone mark reader domains, including the ADD, CHROMO, MBT, PHD, and WD40 domains (Musselman et al, 2012). The TUDOR domain of SMN (SMN_TUDOR) is well known to interact not only with arginine-methylated (R^me) proteins, such as COILIN (Boisvert et al, 2002; Hebert et al, 2002), AVEN (Thandapani et al, 2015), and RNA polII (Zhao et al, 2016), but also with RGG motif-containing proteins independently of R^me, such as FIBRILLARIN (Pellizzoni et al, 2001; Whitehead et al, 2002). Interestingly, in SMA cases involving *SMN1* mutations, alterations congregate within either the TUDOR domain or the tyrosine- and glycine-rich (YG-rich) oligomerization domain (reviewed in Lomonte et al [2020]), suggesting that SMN oligomerization and TUDOR-mediated protein–protein interactions are biologically implicated in the SMA pathology. Although the YG-rich carboxy-terminal region of SMN is essential for oligomerization (Lorson et al, 1998; Martin et al, 2012), the amino-terminal region (i.e., exon 2) mediates protein-protein interaction with GEMIN2 (Liu et al, 1997; Sarachan et al, 2012) and nucleic acids (Lorson et al, 1998).

Conventional co-immunoprecipitation experiments followed by mass spectrometry (MS)-based proteomic analyses have previously identified core SMN-interacting partners (Fuller et al, 2010; Shafey et al, 2010). The BioID proteomic approach is well established, broadly used, and relevant (Go et al, 2021; May et al, 2022). We have thus used the BioID proximity biotinylation approach relying on the fusion of SMN to a mutated form of the biotin ligase BirA (Roux et al, 2012) to expand the repertoire of the SMN-associated proteome and identify factors that may come into SMN vicinity or contact SMN

---

[1]Université Claude Bernard Lyon 1, CNRS UMR 5261, INSERM U1315, LabEx DEV2CAN, Institut NeuroMyoGène-Pathophysiology and Genetics of Neuron and Muscle, Team Chromatin Dynamics, Nuclear Domains, Virus, Lyon, France   [2]University of Ottawa, Faculty of Medicine, Department of Cellular and Molecular Medicine, Ottawa, Canada   [3]Université Grenoble Alpes, INSERM, CEA, UMR BioSanté U1292, CNRS, CEA, FR2048, Grenoble, France   [4]Université de Montpellier, CNRS UMR 9002, Institut de Génétique Humaine, Montpellier, France

Correspondence: olivier.binda@mail.mcgill.ca; patrick.lomonte@univ-lyon1.fr

transiently to regulate its functions, but not necessarily associate physically with SMN. Herein, we have established the BioID approach in a general cellular model to identify novel SMN-interacting partners that may contribute to the severity of the SMA pathology. We have identified well-known SMN partners (e.g., COILIN, EWSR, and GEMIN2-8), thus validating our approach, and poorly characterized partners (e.g., the fragile X mental retardation protein; FMRP [Piazzon et al, 2008]). In addition, we have identified novel partners (e.g., CAPRIN1, eIF4E2, FXR1-2, and GIGYF1) and potentially new regulators (i.e., the protein arginine methyltransferase PRMT1). The BioID approach allowed us to identify FMRP and fragile X-related proteins 1 (FXR1) and 2 (FXR2) as candidates potentially relevant to the severity of SMA. We have then validated the interactions between SMN and fragile X mental retardation protein family members (FMR$_{FM}$). Furthermore, the silencing of the methyltransferase PRMT1 enhanced SMN-FMRP interactions, thus identifying a potential signalling pathway involving R$^{me}$ in the regulation of SMN and FMRP cellular functions in neurodegenerative disorders. Finally, we observed not only that SMN forms droplets in vitro, which are stabilized by RNA, but also that SMN and FMRP co-fractionate with polysomes in an RNA-dependent manner. These findings suggest that SMN may be involved in the formation of membraneless organelles, such as Cajal nuclear bodies, via phase separation.

## Results

### Establishing a model system to identify SMA-relevant SMN-interacting proteins

We have initially established and extensively validated our BirA-SMN proximity labelling system in HEK293T cells, which are immortal transformed human cell lines, but have interestingly neuron-like features, such as morphology and transcriptome (Shaw et al, 2002; Stepanenko & Dmitrenko, 2015). The cells were transfected with a panel of MYC-tagged controls including BirA alone, BirA$_{NLS}$, BirA-SUMO2, BirA-ING3 (an H3K4$^{me3}$ histone mark reader [Kim et al, 2016; McClurg et al, 2018] unrelated to SMN$_{TUDOR}$ as a reader of H3), or BirA-SMN and BirA-SMN mutant forms. After the doxycycline induction of MYC-BirA constructs (or simply BirA hereafter for simplicity), biotin was added to the medium for 24 h. First, we assessed BirA-SMN expression compared with endogenous SMN levels and found that BirA-SMN was expressed less than endogenous SMN (Fig S1A), whereas both BirA-SMN and BirA-SMN$_{Y109C}$ co-localized with the Cajal nuclear body marker COILIN (Fig S1B and C). Then, biotinylated proteins were affinity-purified using streptavidin-Sepharose and analyzed by immunoblotting. We confirmed that known SMN-binding proteins (i.e., COILIN, GEMIN2) are specifically biotinylated by WT BirA-SMN (Fig 1A). Importantly, BirA, BirA-SUMO2, and BirA-ING3 controls failed to appreciably biotinylate GEMIN2 and COILIN to levels as high as BirA-SMN (Fig 1A). Interestingly, SMA-linked SMN$_{TUDOR}$ mutants (SMN$_{ST}$) Y109C and E134K minimally biotinylated COILIN, whereas the aromatic cage Y109A/Y127A/Y130A triple mutant SMN$_{3YA}$ modified COILIN below the background level (Fig 1A). Conversely, all SMN$_{TUDOR}$ mutants retained an efficient capacity to biotinylate GEMIN2. Thus,

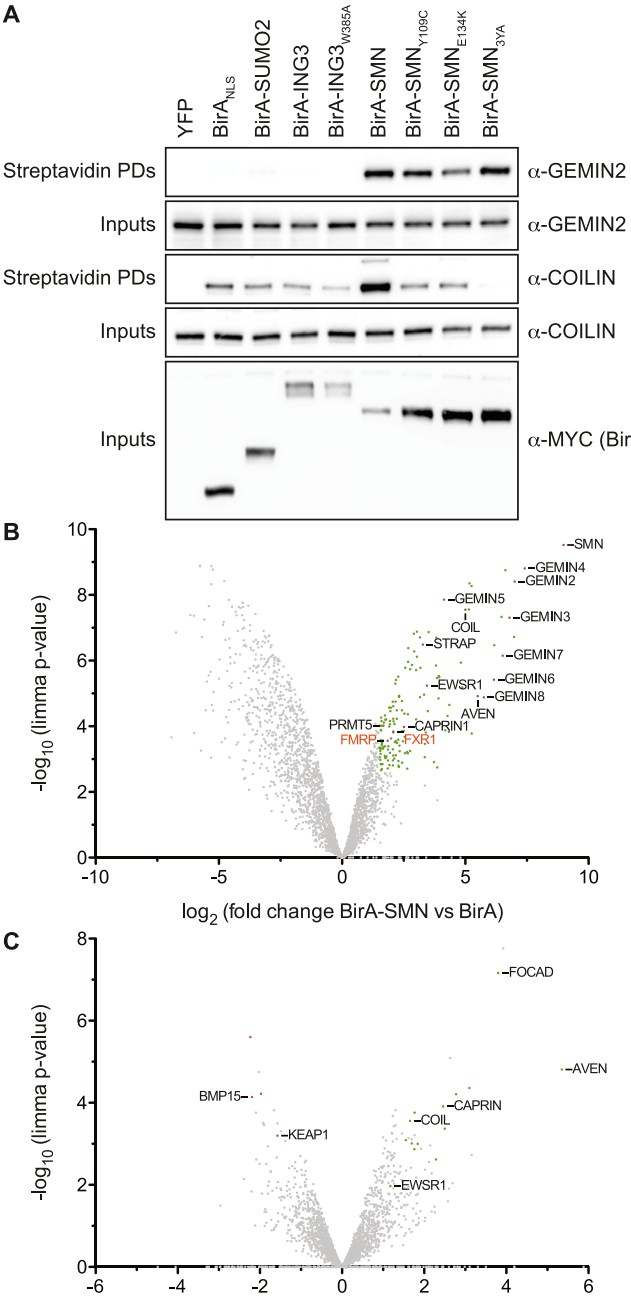

**Figure 1. Establishment of the BirA biotinylation system.**
**(A)** HEK293T cells were transfected with the indicated MYC-tagged BirA constructs. After 24 h, BirA expression was induced with doxycycline at 1 µg/ml in the presence of biotin at 50 µM for an additional 24 h. Cells were washed with PBS to remove excess biotin and lysed. Biotinylated proteins were pulled down with streptavidin-Sepharose beads and analyzed by immunoblotting (n = 4). **(B)** Volcano plot representation of the SMN proxisome. Known SMN-associated proteins are highlighted in black font. **(C)** As in panel (B), but the proxisome of BirA-SMN was compared with BirA-SMN$_{Y109C}$.

mutations expected to impact R$^{me}$ interactions impaired the biotinylation of COILIN (via TUDOR), but not GEMIN2 (via N-terminus). Overall, we conclude that BirA-SMN specifically and reproducibly biotinylates established SMN-associated proteins.

## Defining the SMN proximity proteome (proxisome)

To map without a priori the in cellulo proxisome of SMN, BioID experiments using BirA-SMN and BirA alone were performed in triplicate before MS-based label-free quantitative proteomic analyses. This strategy identified a group of 150 proteins significantly enriched in BirA-SMN eluates compared with BirA eluates (fold change ≥1.5 and $P$ ≤0.0025, allowing to reach a Benjamini–Hochberg false discovery rate [FDR] <1%; Table S1). Among them, as expected, GEMIN2-8 and many classic SMN interactors were identified, such as COILIN, EWSR1, STRAP, FIBRILLARIN, and FUS (Table S1 and Fig 1B; see also STRING analysis [Fig S1D] for a more detailed visual representation). Interestingly, AVEN, an $R^{me}GG$-modified RNA-binding and G-quadruplex–binding protein, was found among the most enriched proteins with SMN, in agreement with the recent work (Thandapani et al, 2015). Bioinformatic analyses aiming at exploring GO terms over-represented in the group of proteins found enriched in the SMN proxisome showed that, as expected, a part of these proteins are members of the SMN complex and associated complexes (i.e., snRNPs and methylosome) and mainly involved in the spliceosomal snRNP assembly (Table S2). Furthermore, these analyses pointed to a group of proteins involved in the negative regulation of translation and include notably AGO1, AGO2, CAPRIN1, eIF4E, eIF4E2, FMRP, GIGYF1-2, and ZNF598 (Iwasaki et al, 2009; Morita et al, 2012; Kim et al, 2019; Weber et al, 2020). We conclude that this proximity proteomic approach not only identified already known partners of SMN, but also expanded the repertoire of its potential associated proteins.

The proxisome of SMN was then extended to include SMA-linked SMN_{TUDOR} mutant form SMN_{Y109C}. A comparison between BirA-SMN and BirA-SMN_{Y109C} proxisomes highlights that the SMA-linked mutation impacted negatively on the association of SMN with factors such as AVEN, CAPRIN1, COIL, FOCAD, GIGYF1, PATL1, and WRAP53 (Fig 1C and Table S1). Interestingly, SMN_{Y109C} seemed to enhance the abundance of proteins such as BMP15, KEAP1, and PGAM5, whereas FMRP and FXR1 were only modestly impacted (Fig 1C and Table S1).

## Validation of BirA-SMN–mediated biotinylation and identification of novel SMN interactants

So far, we have identified factors that may come in proximity to BirA-SMN under conditions that allow their biotinylation. We thus aimed at determining by co-immunoprecipitation whether these candidate partners can actually associate with SMN. To validate novel SMN-interacting candidates, *FMR1* and *FXR1* cDNAs were cloned into a FLAG-tag expressing vector. As *TIRR* did not appear in the HEK293T SMNome and *CSRP2* was found below the fixed cut-off to be considered as enriched with BirA-SMN, these were used as negative controls. The cDNAs were then co-expressed in HEK293T cells with MYC-tagged BirA or BirA-SMN. After induction with biotin and doxycycline, biotinylated proteins were affinity-purified with a streptavidin-Sepharose matrix and analyzed by immunoblotting. Although we did not obtain increased levels of biotinylation on FLAG-CSRP2 or FLAG-TIRR with BirA-SMN compared with BirA alone, we did observe a modest increase in biotinylation on FLAG-FMRP and FLAG-FXR1, consistent with the limited but significant enrichment of these proteins in BirA-SMN proxisome

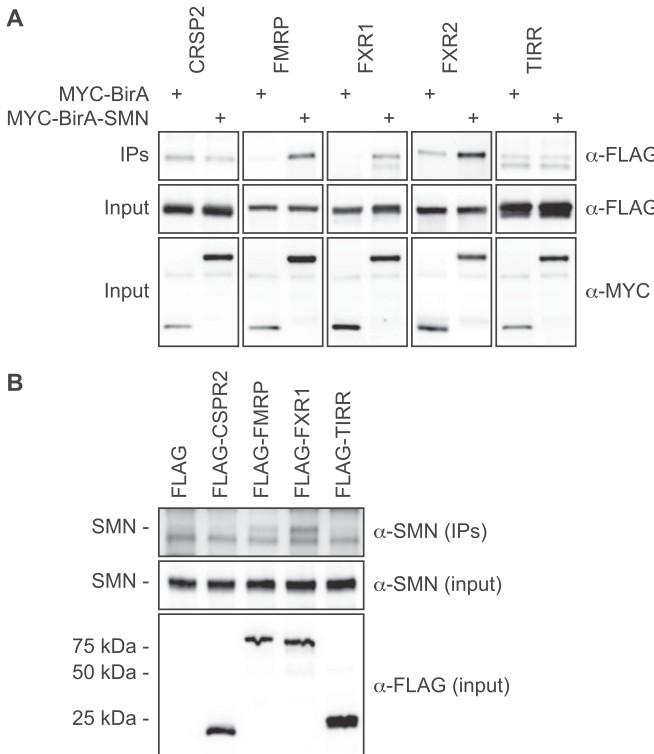

**Figure 2. SMN associates with biotinylated candidates.**
**(A)** FLAG-tagged candidates (i.e., CSRP2, FMRP, FXR1, FXR2, and TIRR) were co-expressed in HEK293T cells with either MYC-BirA or MYC-BirA-SMN. BirA proteins were immunoprecipitated using an α-MYC antibody and immunoprecipitates analyzed by immunoblotting using α-FLAG. Input protein levels were assessed with indicated antibodies. **(B)** As in panel (A), but FLAG-tagged candidates (i.e., CSRP2, FMRP, FXR1, and TIRR) were expressed on their own. Immunoprecipitates were analyzed with α-SMN to detect the presence of endogenous SMN. The upper band is endogenous SMN (highlighted as SMN -). The lower band is an unknown non-specific signal.

compared with BirA alone (Fig S2). This suggests that SMN interacts with those factors or at least comes within close proximity.

We similarly co-expressed the proteins in cells and performed immunoprecipitation using an α-MYC antibody followed by immunoblotting analyses. In agreement with the biotinylation experiments (Fig S2), not only FLAG-FMRP and FLAG-FXR1 but also FLAG-FXR2 (found enriched in the MYC-BirA-SMN proxisome, but below the fixed significance cut-off) co-immunoprecipitated with MYC-BirA-SMN, but not with BirA alone, whereas FLAG-CSRP2 and FLAG-TIRR failed to co-immunoprecipitate in either condition (Fig 2A). To further validate the interaction between endogenous SMN and FLAG-tagged FMR_{FM}, we performed immunoprecipitation using an α-FLAG antibody and revealed the interactions by immunoblotting using an α-SMN antibody. These experiments demonstrate that FMR_{FM} interact with endogenous SMN (Fig 2B), revealing FMRP, FXR1, and FXR2 as genuine SMN-interacting proteins.

## SMN interacts directly with FMRP and FXR1

Interestingly, SMN, FMRP, FXR1, and FXR2 all have TUDOR domains that can facilitate interactions with methylated arginines ($R^{me}$)

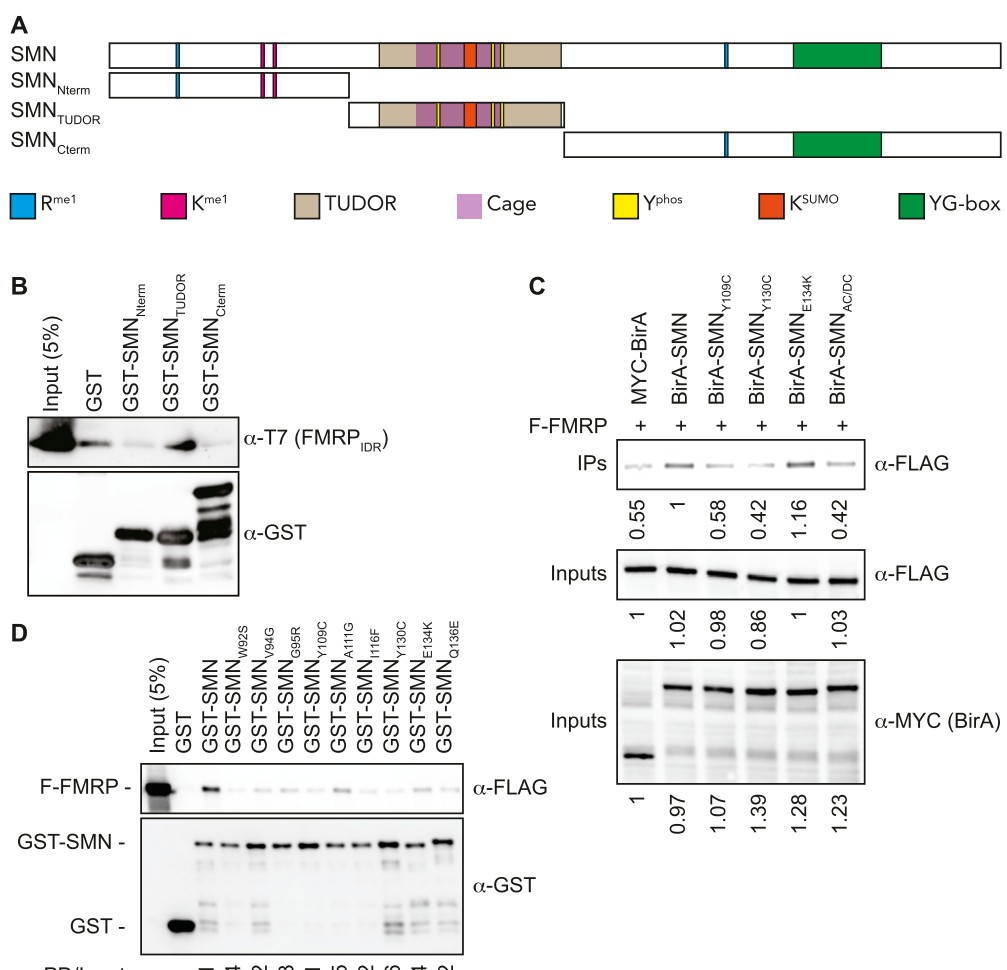

**Figure 3. SMN$_{TUDOR}$ mutants affect the SMN-FMRP interaction.**
**(A)** Graphical representation of full-length WT (SMN$_{WT}$), amino-terminal (SMN$_{Nterm}$), central TUDOR (SMN$_{TUDOR}$), and carboxy-terminal (SMN$_{Cterm}$) regions of SMN protein. **(B)** GST pulldown assays were performed with GST alone or with GST-tagged recombinant SMN truncations along with affinity-purified recombinant FMRP$_{IDR}$, then analyzed with indicated antibodies. **(C)** Either MYC-BirA or MYC-BirA-SMN forms were co-expressed with FLAG-FMRP (F-FMRP) in HEK293T cells. α-MYC IPs were analyzed by immunoblotting using α-FLAG antibody and input protein levels assessed with indicated antibodies. **(D)** Recombinant GST-tagged SMN and SMN$_{ST}$ forms were expressed in *Escherichia coli* and affinity-purified using a glutathione-Sepharose purification scheme. Recombinant proteins were incubated with cell extracts containing FLAG-FMRP (labelled F-FMRP) and pulled down (PD) using glutathione-Sepharose beads. PDs were analyzed by immunoblotting with either α-FLAG or α-GST antibodies. Ratios below panel (B) are between pulled down GST-SMN and pulled down FLAG-FMRP.

(Adams-Cioaba et al, 2010). Although SMN bears a few R$^{me}$ modifications (Guo et al, 2014; Larsen et al, 2016), both FMRP and FXR1 harbour several RGG repeats that are methylated (Stetler et al, 2006). To address which region of SMN associates with FMRP, GST-tagged SMN truncations were generated (represented in Fig 3A). Only the RGG-containing region of FMRP (amino acid residues 445–590 [FMRP$_{IDR}$]) could be expressed and purified. As expected, recombinant SMN$_{TUDOR}$ could on its own associate directly with purified T7-tagged FMRP$_{IDR}$ (Fig 3B). To further explore how SMN$_{TUDOR}$ is involved in mediating interactions with FMRP and FXR1, we co-expressed WT FMR$_{FM}$ with a panel of several SMA-linked TUDOR mutants (i.e., Y109C, Y130C, and E134K) and with an aromatic cage dead construct (i.e., W102A/Y109A/Y127A/Y130A quadruple mutant or SMN$_{AC/DC}$) and performed immunoprecipitation. Interestingly, with the exception of SMN$_{E134K}$, SMN$_{ST}$ mutants and SMN$_{AC/DC}$ failed to immunoprecipitate FLAG-tagged FMRP (Fig 3C) and FXR1 (Fig S3A), suggesting that SMN may interact with these factors via R$^{me}$-dependent protein-protein interactions or otherwise with RGG motifs. To further explore the role of SMN$_{TUDOR}$ in mediating interactions with FMRP and FXR1, we used an extended panel of SMN$_{ST}$ mutants (compiled in Lomonte et al [2020]). These recombinant proteins were expressed in bacteria and affinity-

purified, then used to pull down FLAG-tagged FMRP and FXR1 from cell extracts. Interestingly, all SMN$_{ST}$ mutants impacted, to various degrees, negatively on SMN-FMRP (Fig 3D) and SMN-FXR1 (Fig S3B) interactions, further not only demonstrating that SMN interacts with FMRP and FXR1, but also confirming that an intact SMN$_{TUDOR}$ domain is essential for these interactions.

**SMN-FMRP interaction is negatively regulated by the arginine methyltransferase PRMT1**

Having determined that SMN$_{TUDOR}$ integrity is essential for interactions with FMR$_{FM}$, we aimed to address whether arginine methyltransferases (PRMTs) in general are involved in regulating SMN-FMR$_{FM}$ interactions. Because PRMT5 was found to be significantly enriched in the SMN proxisome, whereas PRMT1 and PRMT3 were found enriched in the SMN proxisome, but just below the fixed significance cut-off (Fig 1B), PRMTs were thus silenced individually using previously validated siRNA pools (Sabra et al, 2013) and interactions of endogenous SMN with FLAG-tagged FMRP were further investigated. Surprisingly, the silencing of PRMT1, unlike PRMT3 or PRMT5, actually enhanced the SMN interaction with FLAG-FMRP (Fig 4A and B), suggesting an R$^{me}$-

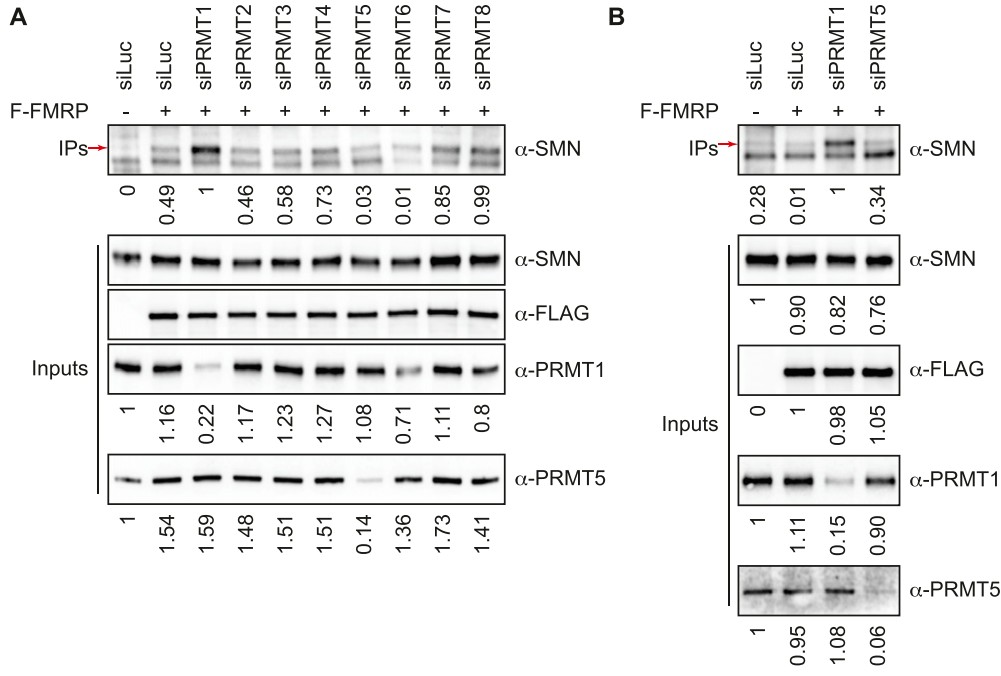

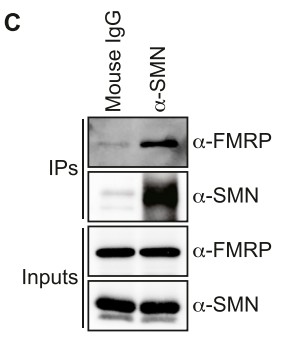

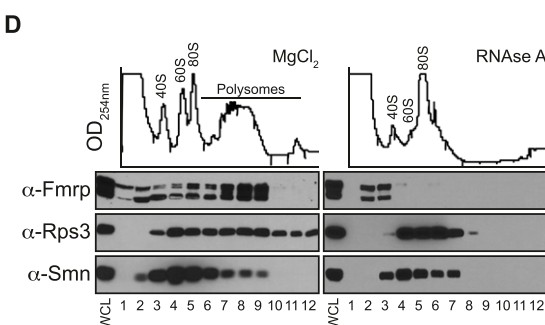

**Figure 4. Silencing of PRMT1 enhances SMN-FMRP interaction.**
**(A)** Expression of arginine methyltransferases (PRMT1-8) was silenced using siRNA pools by reverse transfection in HEK293T cells, which were transfected with FLAG-FMRP (labelled F-FMRP) the next day. $\alpha$-FLAG IPs were analyzed by immunoblotting using $\alpha$-SMN. The red arrow ($\rightarrow$) indicates the endogenous SMN. Input protein levels of SMN and FLAG-FMRP were assessed with $\alpha$-SMN and $\alpha$-FLAG, respectively. Silencing efficiency was assessed for PRMT1 and PRMT5 (bottom panels). **(B)** As in panel (A), but only PRMT1 and PRMT5 were silenced. **(C)** HEK293T cells were cultured on a larger scale, proteins were extracted, and co-immunoprecipitation assays were performed on endogenous proteins. **(D)** Polysome fractions were analyzed by sucrose gradient from mouse MN1 cholinergic motor neuron cell extracts. On the left, MgCl$_2$-treated (control), and on the right, RNase A-treated (100 $\mu$g/ml) samples. Whole-cell extracts were loaded in the first lane.

independent interaction, in agreement with the direct interaction between recombinant SMN$_{TUDOR}$ and RGG-containing FMRP$_{IDR}$ (Fig 3B).

Endogenous SMN and FMRP associate in cells and co-fractionate with polysomes. Finally, we assessed whether endogenous forms of both proteins could associate together. HEK293T cell extracts were subjected to immunoprecipitation with either mouse IgG (negative control) or $\alpha$-SMN antibodies, and we observed that endogenous SMN could co-immunoprecipitate with endogenous FMRP (Fig 4C), validating the proximity proteome (Fig 1B) and various pulldown assays (Figs 2–4).

FMRP is a well-established translation regulator (Liu et al, 2018; Richter & Zhao, 2021). Thus, to explore the hypothesis that SMN and FMRP act together to regulate translation, we performed sucrose fractionation assays to isolate ribosomes from mouse MN1 cholinergic motor neuron cells. We observed a co-fractionation of Smn and Fmrp (Fig 4D, left). Interestingly, upon RNase A treatment, Fmrp shifted to lighter fractions than Smn (Fig 4D, right). Ribosomal protein Rps3 was used as a control.

Overall, we observe that although SMN$_{TUDOR}$ integrity is essential for interactions with FMR$_{FM}$, SMN-FMR$_{FM}$ interactions appear to be impaired by R$^{me}$ mediated by PRMT1 or at least require the absence of the arginine methyltransferase. We thus conclude that R$^{me}$ signalling via PRMT1 regulates SMN-FMR$_{FM}$ interactions, thus likely impacting their functions. Furthermore, endogenous SMN and FMRP co-immunoprecipitate, whereas Smn and Fmrp co-fractionate with MN1 cholinergic motor neuron cell polysomes, suggesting a functional interaction possibly involving translational regulation.

## SMN forms droplets in vitro reminiscent of phase separation

Outside of the TUDOR domain and the YG-rich region, SMN is greatly disordered (Lomonte et al, 2020) (Fig S4A). Notably, there is a short region (amino acid residues 195-248) at the carboxy-terminus that contains 54% proline residues and is not surprisingly predicted to be an intrinsically disordered region (SMN$_{IDR}$). Indeed, the AlphaFold-derived structure (Jumper et al, 2021; Tunyasuvunakool et al, 2021) of full-length SMN highlights three unstructured regions (Fig S4B). Proteins with an IDR, such as FMRP and CAPRIN1, among others, commonly phase-separate (Kim et al, 2019; Garaizar et al, 2020; Tsang et al, 2020; Borcherds et al, 2021). In addition, the compound biotin-isoxazole is found to precipitate phase-separating RNA-binding IDR-containing proteins (Han et al, 2012;

Kato et al, 2012; Terlecki-Zaniewicz et al, 2021). In agreement, we found that isoxazole precipitates TDP43 and SMN in a dose-dependent manner (Fig S4C), further suggesting that SMN may phase-separate.

To investigate the capability of SMN to phase-separate in vitro, the recombinant SMN (rSMN) was purified and concentrated. Upon storage on ice (or at 4°C), SMN preparations promptly become hazy and form a dense viscous precipitate within a few hours, which can be reversed by warming (Fig S5A). Strikingly, under the microscope, rSMN forms droplets that are able to fuse or combine, but that rapidly diffuse (Fig S5B). Given that SMN associates with RNA (Lorson et al, 1998) and RNA is reported to "catalyse" phase separation (Wiedner & Giudice, 2021), total RNA was added to rSMN preparations. Surprisingly, rSMN formed droplets, but these were stabilized by RNA over time at room temperature (Fig 5A and Video 1). Moreover, sub-compartments within rSMN droplets appeared and seemed very dynamic. We thus labelled rSMN with Cy3 and total RNA with fluorescein for further investigations and visualization. Strikingly, the fluorescein-labelled RNA signal overlapped almost completely with the Cy3-SMN signal (Fig 5B). To further characterize SMN droplets, turbidity assays were performed. As observed under the microscope, SMN droplets are stabilized in the presence of total RNA compared with control samples without RNA (Fig 5C). Furthermore, RNase A pre-treatment of the RNA led to reduced turbidity (absorbance at $OD_{330nm}$) (Fig 5C), highlighting the importance of RNA in the maintenance of SMN droplets over time.

As an additional control experiment, Cy5-labelled recombinant HP1α (not found in the SMN proxisome and available in the laboratory) was added to the mixture and observed to be excluded from SMN droplets (Fig 5D). In contrast, the Cy5-labelled $FMRP_{IDR}$ signal overlapped broadly with Cy3-SMN and fluorescein-labelled RNA signals (Fig 5E), as expected from an SMN-interacting partner (Fig 3B). We thus conclude that rSMN forms droplets, which are stabilized by RNA and can encapsulate direct interaction partners.

### SMA-linked SMN mutants prevent droplet formation

Given that the carboxy-terminus of SMN mediates oligomerization (Martin et al, 2012), we hypothesized that the oligomerization of SMN may allow local SMN concentration to increase and thus promote droplet formation. To investigate this possibility, we generated the SMA-linked G275S mutant, which is reported to remain in a monomeric form (Martin et al, 2012). Interestingly, the Cy5-labelled $SMN_{G275S}$ did not form droplets whether in the presence or absence of RNA and did not appear to overlap with Cy3-labelled SMN droplets (Fig 6A), but did form aggregates in neighbouring fields (Fig 6B). To more broadly investigate the impact of SMA-linked SMN mutations on the formation of droplets, the $SMN_{E134K}$ mutant was generated. Although the $SMN_{E134K}$ signal overlapped with fluorescein-labelled RNA, it was found to aggregate instead of forming droplets (Fig 6C). Then, the $SMN_{ST}$ Y109C and Y130C mutants were also expressed and purified. Again, $SMN_{Y109C}$ and $SMN_{Y130C}$ signals overlapped with RNA, but appeared as aggregates instead of droplets (Fig S6A). Finally, we verified that the fluorescein label on its own did not localize with Cy3-labelled SMN droplets. Indeed, only fluorescein-labelled RNA

signal overlapped with SMN droplets, not free fluorescein dye (Fig S6C and D). We thus conclude that SMN requires an intact TUDOR domain, RNA, and oligomerization potential to self-associate and form droplets, whereas SMA mutant forms of SMN aggregate.

## Discussion

Previous proteomic analyses of SMN-associated proteins focussed on WT SMN and relied on conventional immunoprecipitation (Fuller et al, 2010; Shafey et al, 2010). Herein, we took advantage of a proximity-based biotin ligase approach combined with SMA-linked SMN mutants to identify new and potentially pathologically relevant SMN-interacting candidates. As expected, we identified several known SMN partners, such as GEMIN2-8. Importantly, we identified numerous new candidates that were biotinylated by BirA-SMN, but not by BirA-SMN mutants. Given the broad role of SMN in RNA metabolism, it was encouraging to find that several candidates are involved in RNA binding and are playing major roles in various steps of RNA metabolism, notably mRNA splicing (Table S2).

Moreover, FMRP was previously found in Cajal nuclear bodies (Dury et al, 2013), and although it was not found in previous SMN interactomic studies (Fuller et al, 2010; Shafey et al, 2010) or listed in GenBank, a previous study suggested that SMN and FMRP are interacting together (Piazzon et al, 2008). However, the functional role of a potential SMN-FMRP interaction was never determined (Piazzon et al, 2008), thus warranting further investigations. Precisely, although SMN was shown over a decade ago to associate with FMRP (Piazzon et al, 2008), nothing is known about the cellular function of the SMN-FMRP interaction. Herein, we found that FMRP and SMN co-fractionate with polysomes in an RNA-dependent manner, suggesting a role in translation regulation. In addition to FMRP, we discovered that SMN also associates with $FMR_{FM}$ FXR1 and FXR2. Moreover, we found that SMA-linked TUDOR mutant forms ($SMN_{ST}$) failed to associate with $FMR_{FM}$. Thus, we conclude that an intact TUDOR domain is required for interactions between SMN and $FMR_{FM}$ (Figs 3 and S3).

Interestingly, the silencing of the arginine methyltransferase PRMT1 enhanced SMN-FMRP interactions (Fig 4), suggesting that arginine methylation ($R^{me}$) is detrimental to SMN-FMRP interactions. Notably, arginine methylation of FMRP by PRMT1 prevents the association with G-quadruplex RNA (Blackwell et al, 2010), reminiscent of the modulation of SMN-FMRP interactions by PRMT1 (Fig 4). Together with our current study, these results suggest that PRMT1 could regulate the association of SMN with $FMR_{FM}$, which could work together to resolve G-quadruplex RNA. Furthermore, PRMT1 methylates RGG motifs within an intrinsically disordered region of FMRP involved in phase separation, thus suggesting that phase separation may dictate how or when SMN interacts with FMRP. Moreover, $FMRP_{R me GG}$ regulates association with certain RNA molecules and polyribosomes (Blackwell et al, 2010), whereas PRMT1-catalysed $R^{me}$ prevents FMRP from phase separation and inhibits translation (Tsang et al, 2019). These aspects would be fascinating to investigate in future.

Another TUDOR domain protein, TDRD3, associates with $R^{me}$ peptides (Côté & Richard, 2005). Like SMN, TDRD3 also associates

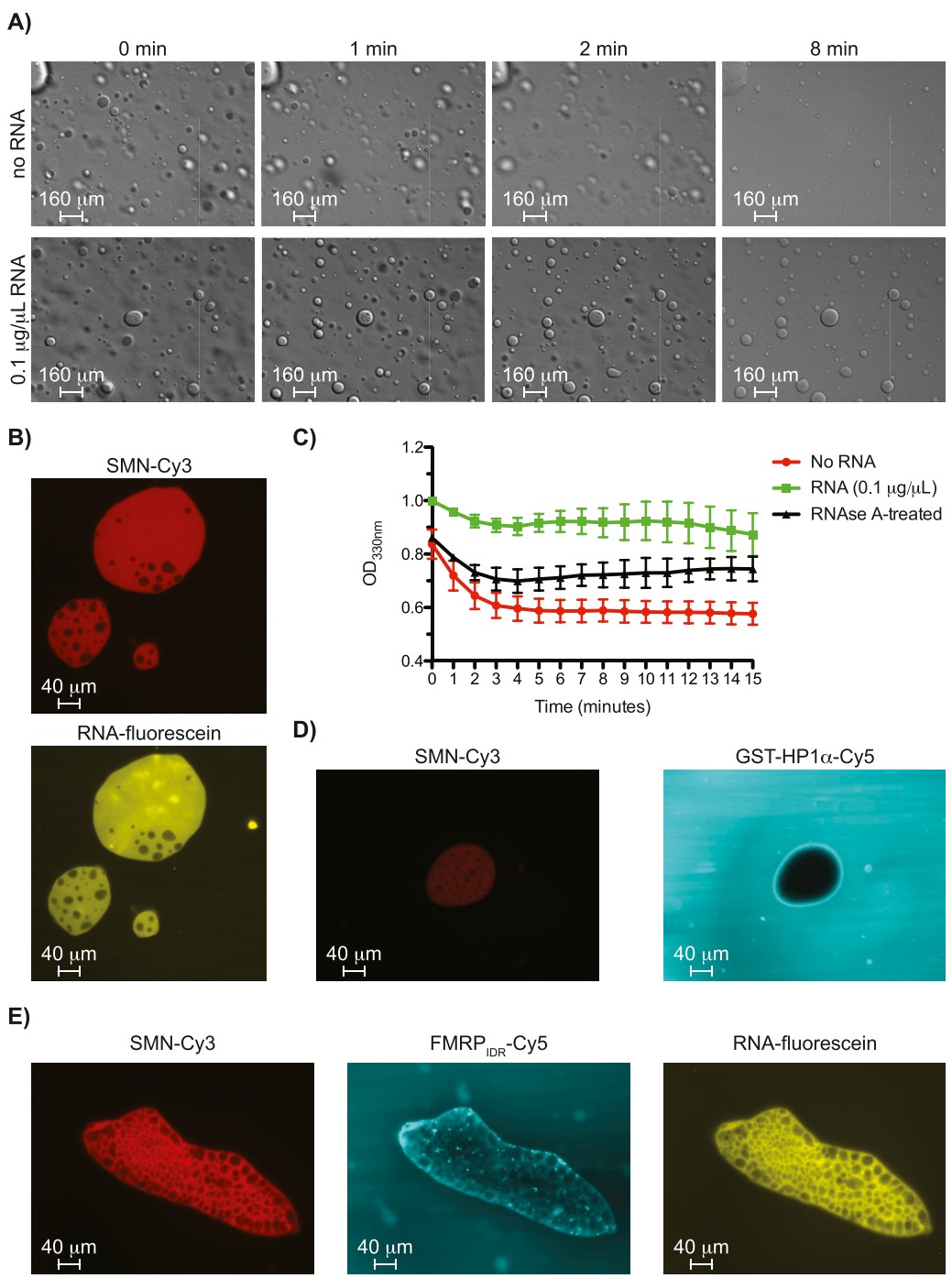

**Figure 5.  RNA stabilizes SMN droplets.**
**(A)** SMN forms droplets in vitro that are reminiscent of phase separation. These droplets diffuse rapidly over time at room temperature (top row of panels). In the presence of RNA (lower set of panels), SMN droplets persist longer. **(B)** rSMN was labelled with Cy3 and total RNA from HEK293T cells labelled with fluorescein for visualization. Droplets of rSMN were immobilized under a coverslip for image capture at 40× magnification. **(C)** As in panel (B), but turbidity was assessed by optical density at 330 nm (OD$_{330}$) on a Synergy H1 plate reader over time (every minute for 15 min) in triplicate. **(D)** As in panel (B), but Cy5-labelled GST-HP1α was included as a negative control showing exclusion from rSMN droplets. **(E)** As in panel (B), but Cy5-labelled FMRP was added to the droplet mixture. All images were captured with a 40X objective, except panel (A) (10X).

with FMRP and polyribosomes in stress granules (Goulet et al, 2008; Linder et al, 2008). However, the TUDOR domain of TDRD3 is dispensable for the interaction with FMRP (Linder et al, 2008), but required for localization to stress granules (Goulet et al, 2008).

Based on our data with BirA-SMN and BirA-SMN$_{ST}$, we envision that other pathology-associated mutants (e.g., FUS or TDP-43 mutants found in ALS) will be invaluable in a proteomic context to dissect molecularly neurodegenerative disorders.

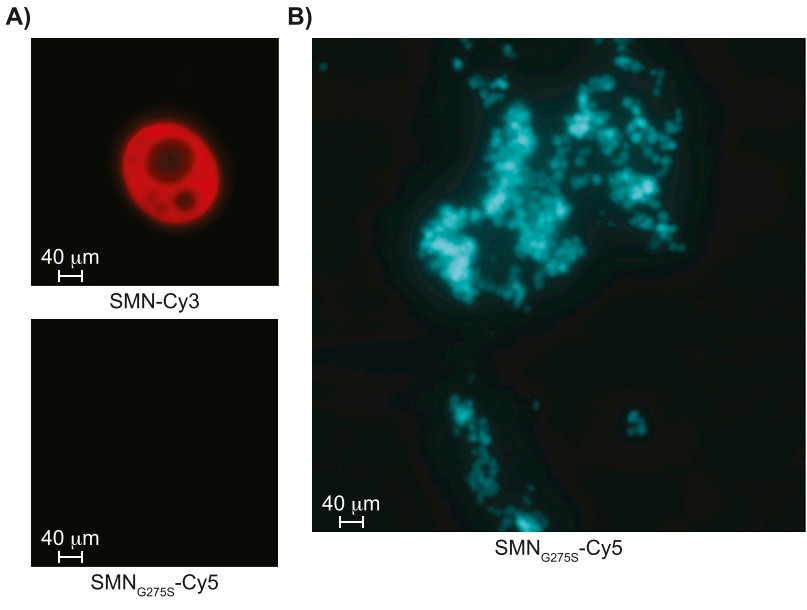

**A)**

SMN-Cy3

SMN$_{G275S}$-Cy5

**B)**

SMN$_{G275S}$-Cy5

**Figure 6. SMA-linked SMN mutants prevent droplet formation.**
**(A)** Recombinant SMN and SMN$_{G275S}$ were labelled with Cy3 and Cy5, respectively, mixed, and allowed to form droplets. Droplets were visualized as described in Fig 5. **(B)** SMN$_{G275S}$ formed aggregates. **(C)** SMN$_{E134K}$ formed aggregates that overlap with fluorescein-labelled RNA. All images were captured with a 40X objective.

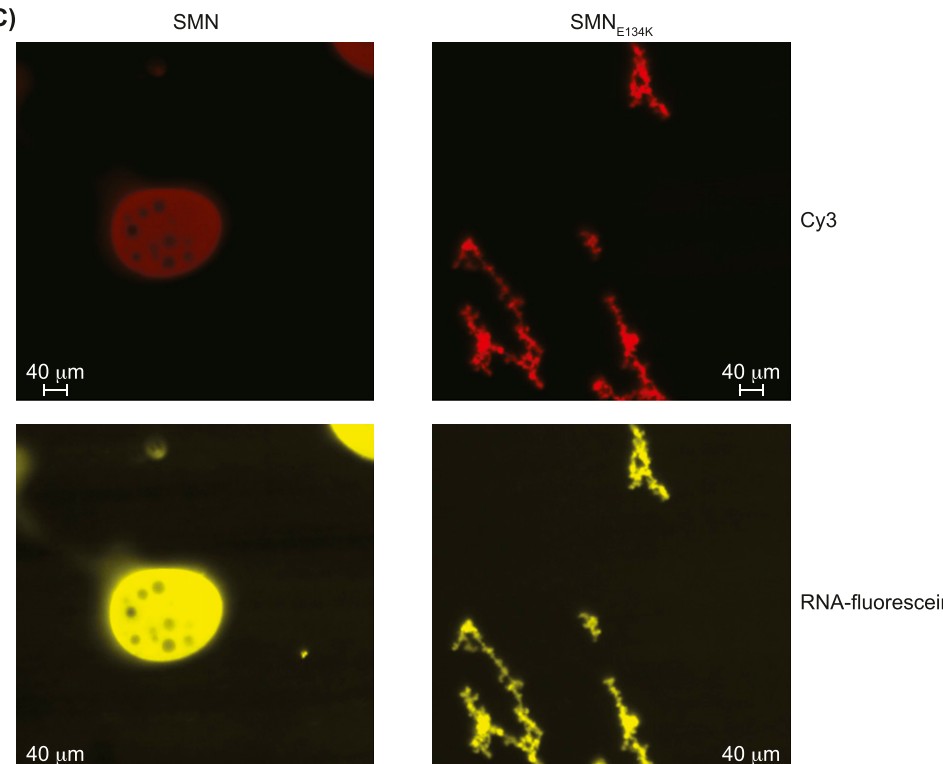

**C)**

SMN | SMN$_{E134K}$ | Cy3

RNA-fluorescein

Given that SMN is capable of nucleating de novo Cajal nuclear bodies (Kaiser et al, 2008), which are membraneless organelles formed via a phase separation mechanism (Riback et al, 2020), and that SMN is sufficient for the formation of droplets in vitro (Fig 5), we suggest that SMN is a good candidate for the formation of Cajal nuclear bodies via phase separation. Indeed, SMN$_{TUDOR}$ was recently shown to mediate phase separation in the presence of R$^{me}$ proteins (Courchaine et al, 2021), but also condense into Cajal bodies through S49 and S63 phosphorylation (Schilling et al, 2021). In agreement, we demonstrate biochemically that SMN forms on its own droplets that are stabilized by RNA (Fig 5A), but requires oligomerization potential and an intact TUDOR domain (Figs 6B and C and S6A), suggesting that multiple signalling pathways would regulate SMN phase separation. Finally, it is interesting to observe that SMN droplet formation is RNA-dependent (at least for stability) and that polysome co-fractionation of FMRP and SMN is also requiring RNA.

# Conclusion

We have identified several new SMN candidate partners that function in RNA metabolism and translation. Notably, the FMR$_{FM}$ were of interest and we showed that R$^{me}$ signalling driven by PRMT1 regulates SMN-FMRP interactions. In addition, we show biochemically for the first time that SMN is sufficient on its own to form droplets, suggesting that it could drive the formation of membraneless organelles, such as Cajal nuclear bodies and stress granules.

# Materials and Methods

### Cloning and plasmids

*SMN1* cDNA was amplified using Turbo Pfu (Stratagene) and inserted into a modified pLVX doxycycline-inducible vector containing a MYC-tagged BirA using pLVX-TetOne (631849; Clontech) and BirA. *SMN1* was also inserted in pGEX-6P1 using *BamHI* and *XhoI*, and in pET28a containing an intein sequence and chitin-binding protein tag (MxE-CBP) using *BamHI* and *NotI*. cDNAs from candidate genes were cloned from the total RNA extracted from GM03813 cells (Coriell Institute) using TRIzol (Invitrogen) after reverse transcription using VILO (Invitrogen) and then inserted into pCMV 3×FLAG (Stratagene) using *BamHI* (New England Biolabs) and *XhoI* (Promega). All constructs were sequence-verified by Sanger sequencing services from Biofidal. The lentiviral packaging plasmids pMD2.G (12259; Addgene) and psPAX2 (12260; Addgene) were provided by Dr Didier Trono. Sequences for primer sets used to amplify cDNA of interest or site-directed mutagenesis can be provided upon request.

### Cell culture, viral production, and transductions

HEK293T cells were cultured in DMEM (Sigma-Aldrich) supplemented with penicillin, streptavidin, and glutamine (Sigma-Aldrich). For lentiviral production, HEK293T cells were seeded at a density of 3,000,000 cells per 100-mm plate. The next day, cells were transfected with 12 $\mu$g of pLVX, pMD2.G, and psPAX2 plasmid DNA by the calcium phosphate method. Supernatants were collected at 48 and 72 h post-transfection, combined, filtered by passing through low-protein-binding 0.45-$\mu$m filters (Millipore), and concentrated with Lenti-X (Clontech). To titrate roughly the productions, different amounts of lentiviral particles were applied to target cells in six-well plates overnight in the presence of 8 $\mu$g/ml polybrene (Sigma-Aldrich). 2 d post-transduction, cells were selected with 0.5 $\mu$g/ml puromycin (InvivoGen).

### Proximity biotinylation assay

Stably transduced HEK293T cells (i.e., BirA, BirA-SMN$_{Y109C}$, and BirA-SMN) were grown to confluence in duplicate 150-mm dishes. The expression of BirA and BirA-SMN was induced for 48 h in total using 1 $\mu$g/ml doxycycline (Sigma-Aldrich). After 24 h, biotin (Sigma-Aldrich) was added at 50 $\mu$M for another 24 h. Cells were rinsed

twice with 5 ml ice-cold PBS to remove excess biotin, which interferes with the streptavidin-Sepharose purification scheme. Cells were harvested by scraping in PBS, resuspended in 600 $\mu$l lysis buffer (50 mM Tris, pH 8.0, 140 mM NaCl, 1 mM EDTA, 10% glycerol, 0.5% NP-40, and 0.25% Triton X-100, supplemented with EDTA-free Complete protease inhibitor cocktail [Roche]), briefly sonicated, and cleared by centrifugation for 10 min at 25,000 rcf. Cleared lysates were incubated for 2 h at 4°C on a rotator with 25 $\mu$l pre-washed streptavidin-Sepharose beads (GE Healthcare). After 2 h of incubation, samples were washed four times with 1 ml lysis buffer and resuspended in 50 $\mu$l Laemmli sample buffer.

### MS-based proteomic analyses

Three replicates of BirA, BirA-SMN, and BirA-SMN$_{Y109C}$ proxisomes were prepared. The eluted proteins solubilized in Laemmli buffer were stacked in the top of a 4-12% NuPAGE gel (Invitrogen). After staining with R-250 Coomassie Blue (Bio-Rad), proteins were digested in-gel using modified trypsin (sequencing purity; Promega), as previously described (Casabona et al, 2013). The resulting peptides were analyzed by online nanoliquid chromatography coupled to MS/MS (UltiMate 3000 RSLCnano and Q-Exactive HF; Thermo Fisher Scientific) using a 120-min gradient. For this purpose, the peptides were sampled on a precolumn (300 $\mu$m × 5 mm PepMap C18; Thermo Fisher Scientific) and separated in a 75 $\mu$m × 250 mm C18 column (Reprosil-Pur 120 C18-AQ, 1.9 $\mu$m, Dr. Maisch). The MS and MS/MS data were acquired by Xcalibur (Thermo Fisher Scientific).

Peptides and proteins were identified by Mascot (version 2.6.0; Matrix Science) through concomitant searches against the UniProt database (*Homo sapiens* taxonomy, June 2020 download), a homemade database containing the sequences of classical contaminant proteins found in proteomic analyses (human keratins, trypsin, etc.), and the corresponding reversed databases. Trypsin/P was chosen as the enzyme, and two missed cleavages were allowed. Precursor and fragment mass error tolerances were set respectively at 10 ppm and 25 mmu. Peptide modifications allowed during the search were as follows: Carbamidomethyl (C, fixed), Acetyl (Protein N-term, variable), Biotin (K, variable), and Oxidation (M, variable). The Proline software (Bouyssié et al, 2020) was used for the compilation, grouping, and filtering of the results (conservation of rank 1 peptides, peptide length ≥6 amino acids, peptide score ≥25, FDR of peptide-spectrum-match identifications <1% as calculated on peptide-spectrum-match scores by employing the reverse database strategy, and minimum of one specific peptide per protein group). Proline was then used to perform a MS1 quantification of the identified protein groups based on razor and specific peptides. The MS proteomic data have been deposited to the ProteomeXchange Consortium via the PRIDE (Perez-Riverol et al, 2019) partner repository with the dataset identifier PXD030970.

Statistical analysis was then performed using the ProStaR software (Wieczorek et al, 2017). Proteins identified in the contaminant database, proteins identified by MS/MS in less than two replicates of one condition, and proteins detected in less than three replicates of one condition were discarded. After log$_2$ transformation, abundance values were normalized by median

centring before missing value imputation (SLSA algorithm for partially observed values in the condition and DetQuantile algorithm for totally absent values in the condition). Statistical testing was then conducted using limma, whereby differentially expressed proteins were sorted out using a $\log_2$ (fold change) cut-off of 1.585 and a $P$-value cut-off of 0.0085, leading to a FDR inferior to 1% according to the Benjamini–Hochberg estimator. Proteins found differentially abundant but identified by MS/MS in less than two replicates and detected in less than three replicates in the condition in which they were found to be more abundant were invalidated ($P$ = 1).

### Bioinformatic analyses

Proteins found significantly enriched in the BirA-SMN proxisome were submitted to statistical over-representation tests in PANTHER (Mi et al, 2121). The enrichment of GO terms in Cellular Component, Biological Process, and Molecular Function instances was validated with statistical over-representation tests in PANTHER if $P \geq 5$ and with a corresponding Bonferroni-corrected Fisher's exact test if $P \leq 0.01$.

### SMN-interacting candidates and proximity validation

HEK293T cells were seeded at 3,000,000 cells per 100-mm dish. The next day, cells were transfected with 12 $\mu$g each of the specified pCMV-FLAG and pLVX-BirA constructs using the calcium phosphate method. After 24 h, the medium was changed and supplemented with 1 $\mu$g/ml doxycycline (Sigma-Aldrich) and 50 $\mu$M biotin (Sigma-Aldrich).

### Immunoprecipitation

Briefly, 48 h after transfection, cells were harvested, resuspended in 600 $\mu$l lysis buffer (50 mM Tris-Cl, pH 7.5, 200 mM NaCl, 0.1% Triton X, and 10% glycerol, supplemented with EDTA-free Complete protease inhibitor cocktail [Roche]), and briefly sonicated. Lysates were cleared by centrifugation (25,000 rcf for 10 min) and then incubated for 2 h with 1 $\mu$g specified antibody. The immuno-complexes were then purified using 5 $\mu$l washed Protein G Dynabeads (Invitrogen) and samples further incubated at 4°C for another 2 h. Beads were then washed four times with 1 ml lysis buffer and resuspended in 20 $\mu$l Laemmli sample buffer.

### Recombinant protein purification

BL21 (DE3) (Stratagene) cells were transformed with pGEX6-P1-SMN. BL21 cells were grown in 250 ml LB until they reached $OD_{600}$ of 0.6, and GST-SMN expression was induced with 0.2 mM IPTG for 2.5 h at 37°C. Cells were harvested by centrifugation, and lysed in lysis buffer composed of 50 mM Tris, pH 7.5, 150 mM NaCl, and 0.05% NP-40, supplemented with EDTA-free Complete protease inhibitor cocktail (Roche) on ice.

For SMN-MxE-CBP purification, cells were induced with 0.2 mM IPTG overnight at 16°C and were lysed in room temperature arginine lysis buffer composed of 50 mM Tris, pH 8.0, 150 mM NaCl, 10% glycerol, 0.05% NP-40, and 250 mM arginine (readjust pH after the addition of Arg), supplemented with EDTA-free

Complete protease inhibitor cocktail (Roche) buffer. SMN was purified using chitin-agarose beads (New England Biolabs), labelled with Cy3-maleimide (Lumiprobe), and released by cleaving the intein moiety with 100 mM DTT (Sigma-Aldrich) all at room temperature. To induce droplet formation, SMN was concentrated on an Amicon 3,000 MWCO column (Millipore) to ~90 $\mu$M and stored at 4°C.

### Microscopy and droplet imaging

On a clean glass slide, a drop of total HEK293T (1 $\mu$l of 0.5 $\mu$g/$\mu$l) RNA was deposited. For RNA visualization, RNA was labelled with fluorescein (Cat. MIR3225; Mirus). Then, 4 $\mu$l of purified rSMN were added and mixed gently by pipetting. All images were taken at 20X or 40X magnification on an AxioImager Z1 microscope.

### Tubidity assays

Droplet formation was induced as in the "Microscopy and droplet imaging" section. Samples were supplemented with 0.1 $\mu$g/$\mu$l total RNA or not. Total RNA was also treated with RNAse A (6.7 $\mu$g/$\mu$l) prior to conducting turbidity assays. Turbidity was assessed by optical density at 330 nm (OD330) on a Synergy H1 plate reader over time (every minute for 15 min) in triplicate.

### Immunofluorescence

HEK293T cells were transfected with pLVX vectors containing MYC-tagged BirA, BirA-SMN, or BirA-SMNY109C using Lipofectamine 3000 (Invitrogen) following the manufacturer's protocol. 24 h after transfection, BirA expression was induced using 1 $\mu$g/ml doxycycline. 24 h after induction, cells were fixed for 10 min at room temperature in 2% PFA. Cells were then permeabilized with 0.5% Triton, diluted in 1× PBS and 3% NGS (normal goat serum), then incubated for 1 h with primary antibody and 1 h with secondary antibody. MYC-tagged BirA, BirA-SMN, and BirA-SMN$_{Y109C}$ were detected using an $\alpha$-MYC primary antibody diluted 1:500 (ab9106; Abcam). COILIN was detected using an $\alpha$-COILIN diluted 1:1,000 (ab11822; Abcam). Endogenous SMN was detected using an $\alpha$-SMN diluted 1:200 (610647; BD Biosciences). Highly cross-adsorbed secondary antibody AF488 $\alpha$-rabbit was diluted 1:1,000 (A32731; Thermo Fisher Scientific), and highly cross-adsorbed secondary antibody AF555 $\alpha$-mouse was diluted 1:1,000 (A32727; Thermo Fisher Scientific). DNA was stained using DAPI. Coverslips were mounted using Fluoromount-G (Invitrogen), and samples were observed using a Z1 Axio Observer (Zeiss) at 1,000X magnification.

### Antibodies

The COILIN (AB1005) is from Dundee Cell Product, $\alpha$-FLAG (F1804) is from Sigma-Aldrich, $\alpha$-GST (ab3416) is from Abcam, $\alpha$-MYC (SC-40) is from Santa Cruz, $\alpha$-FMRP (#4317) is from Cell Signaling Technology, $\alpha$-PRMT1 (07-404) and $\alpha$-PRMT5 (07-405) are from Upstate, and $\alpha$-SMN (610647) is from BD Biosciences.

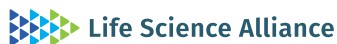

### Sucrose polysome fractionation

MN1 cells were maintained in DMEM, and sucrose fractionation was essentially performed as previously published (Sanchez et al, 2013).

## Data Availability

The MS proteomic data have been deposited to the ProteomeXchange Consortium via the PRIDE partner repository with the dataset identifier PXD030970 (https://www.ebi.ac.uk/pride/).

## Supplementary Information

## Acknowledgements

P Lomonte, O Binda, and F Baklouti are funded by an AFM-Téléthon grant (MyoNeurALP). P Lomonte and O Binda are also funded by the Joint Collaborative Research Program between the Centre for Neuromuscular Disease (University of Ottawa) and Institut NeuroMyoGène (INMG, Claude Bernard Université Lyon 1). The authors acknowledge the Cell Biology and Image Acquisition Core, which is funded by the University of Ottawa (Canada) and the Canada Foundation for Innovation. The proteomic experiments were partially supported by Agence Nationale de la Recherche under projects ProFI (Proteomics French Infrastructure; ANR-10-INBS-08) and GRAL, a Program from the Chemistry Biology Health Graduate School of Université Grenoble Alpes (ANR-17-EURE-0003). J Côté is funded by a Canadian Institutes of Health Research grant (MOP 123381) and CureSMA Canada. P Lomonte is a CNRS research director.

### Author Contributions

O Binda: conceptualization, funding acquisition, investigation, methodology, and writing—original draft, review, and editing.
F Juillard: investigation.
JN Ducassou: investigation.
C Kleijwegt: investigation.
G Paris: investigation.
A Didillon: investigation.
F Baklouti: funding acquisition and writing—review and editing.
A Corpet: supervision and investigation.
Y Couté: supervision, funding acquisition, and investigation.
J Côté: supervision, funding acquisition, and writing—review and editing.
P Lomonte: conceptualization, supervision, funding acquisition, project administration, and writing—original draft, review, and editing.

### Conflict of Interest Statement

The authors declare that they have no conflict of interest.

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
