## [Reviewer comments · Life Science Alliance]

Life Science Alliance

SMA-linked SMN mutants prevent phase separation and SMN interactions with FMRP family members

Olivier Binda, Franceline Juillard, Julia Ducassou, Constance Kleijwegt, Genevieve Paris, Andréanne Didillon, Faouzi Baklouti, Armelle Corpet, Yohann Couté, Jocelyn Côté, and Patrick Lomonte

DOI: <https://doi.org/10.26508/lsa.202201429>

Corresponding author(s): Olivier Binda, University of Ottawa and Patrick Lomonte, French National Centre for Scientific Research

Review Timeline:

Submission Date:	2022-02-28
Editorial Decision:	2022-03-25
Revision Received:	2022-09-06
Editorial Decision:	2022-09-28
Revision Received:	2022-10-15
Editorial Decision:	2022-10-17
Revision Received:	2022-10-24
Accepted:	2022-10-25

Transaction Report:

March 25, 2022

Re: Life Science Alliance manuscript #LSA-2022-01429-T

Dr. Olivier Binda
University of Ottawa
Canada

Dear Dr. Binda,

Thank you for submitting your manuscript entitled "SMA-linked SMN mutants prevent phase separation and SMN interactions with FMRP family members" to Life Science Alliance. The manuscript was assessed by expert reviewers, whose comments are appended to this letter. We invite you to submit a revised manuscript addressing the Reviewer comments.

The typical timeframe for revisions is three months, however if additional time is needed to address the Reviewers' comments, please let me know. Please note that papers are generally considered through only one revision cycle, so strong support from the referees on the revised version is needed for acceptance.

Thank you for this interesting contribution to Life Science Alliance. We are looking forward to receiving your revised manuscript.

Sincerely,

Eric Sawey, PhD
Executive Editor
Life Science Alliance
<http://www.lsa-journal.org>

B. MANUSCRIPT ORGANIZATION AND FORMATTING:

Reviewer #1 (Comments to the Authors (Required)):

In the present manuscript, Binda and colleagues report on the establishment of a Bio-ID strategy to identify novel interaction partners of SMN in human cells. They list and further validate Fragile X mental (FMR) family member proteins as SMN interactors picked up after biotinylation in the vicinity of SMN. These interactions are shown to depend on an intact TUDOR domain in SMN. The authors further investigate a potential function of R-methylation on interaction between SMN and FMR family members using RNAi against a variety of R-methyl transferases. The knockdown of PRMT1 turns out to specifically stimulate interaction. Finally, the authors provide evidence for phase separation of SMN using purified, recombinant SMN as wt and SMA-associated mutant variants. SMN's phase separation seems to be promoted under conditions of co-condensation with RNA species, while mutants in the TUDOR domain negatively impact phase separation in vitro.

Taken together, this is a mixed bag of observations that are, individually mostly well documented and described. The paper is written well and easy to read. However, I failed to find a consistent take home message from the data provided. What is the new insight into SMN function and/or into the molecular basis for SMA? I am listing major concerns, which, from my point of view, preclude publication of the manuscript at this point.

1. The Bio-ID approach is well done in its biochemical and data analysis part. We see biotinylation of SMN complex proteins as "top hits". So far so good. However, a characterization of the cellular situation upon induction of the Bio-ID construct is missing, but would be essential to recapitulate the status of SMN in cells under conditions of biotinylation. What are relative levels of exogenous BirA-SMN to endogenous SMN? How does BirA-SMN localize (Cajal bodies?) and how does the "cloud" of biotinylated substrates develop over time when biotinylation starts?
2. The identification of FMR family proteins, as the authors state themselves in the discussion (see ref 43), as interaction partners of SMN is NOT novel although highlighted as proof of principle for the power of the Bio-ID approach to identify novel SMN interactors. This is at least irritating.
3. The approach to further investigate the dependence of FMR-SMN interactions of "reader" domains (TUDOR in SMN) is a valid one and comes with a good start. However, it is not leading anywhere in the end. What do the interaction and regulation mean? The authors cite ref43 saying that that paper, albeit identifying the interaction of FMR with SMN, did not assign any functional experiments. The present study, unfortunately, does not either. I do appreciate the clear-cut dependence on the TUDOR domain, but this does not elucidate the function any further.
4. The (interesting and clean) observation that PRMT1 knockdown specifically increases SMN-FMR interaction is not taken further but only comes up in the discussion with some rather vague speculations of what this may mean. Is it really the catalytical activity that plays a role here, or the protein itself, as the authors ask themselves but do not further test. Where do PRMT1 and SMN interact (nucleus?) to impact on communication with FMR proteins (cytoplasm?).
5. The paper finally shifts gears and analyses phase separation of recombinant SMN. The stabilization of condensates via RNA is intriguing although the experiment is neither carefully quantified (single images without scale bars...) nor controlled, e.g. via an RNase treatment, using fluorescein alone or using SMN variants deficient in RNA binding.

Minor issues:

Figure 1A: how can a stronger interaction of Bir-NLS with coilin than SMN mutants be explained? In other words: are there (and why) different backgrounds in interaction in this experiment?

Figure 1B: Using a TUDOR domain mutant to compare with wt SMN would have been very insightful...

Figure 2B: The SMN blot looks clean in inputs but not in the IP fractions, why is that?

Figure 4A and B: see 2B

Reviewer #2 (Comments to the Authors (Required)):

In this paper entitled "SMA-linked SMN mutants prevent phase separation and SMN interactions with FMRP family members", Binda and colleagues have investigated by proximity labelling the interactome of the RNA binding protein (RBP) SMN1, a protein causative of the neuromuscular disease SMA. The authors have found that the RBPs FMRP, FXR1 and FXR2 interact with SMN1 and this binding is impaired in the absence of the methyltransferase PRMT1. Moreover, SMN forms RNA-containing liquid droplets in vitro, suggesting that SMN LLPS could be involved in the formation of cajal bodies within cells, and mutations in SMN

lead to the formation of aggregate-like structures.

The topic investigated is of high interest, however the data, as presented in this manuscript, is very preliminary and the experiments presented do not always support the conclusions. A better characterisation of the system is needed, moreover, thorough validations and controls, as well as quantifications and n numbers are missing throughout the manuscript. For these reasons this manuscript should not be accepted in its current form.

Main points:

- Alterations in SMN1 cause alterations in motor neurons, ultimately leading to the neuromuscular disease SMA. Investigating how mutations in SMN1 alter its interactome is definitely of interest, however the expression of disease-relevant RBPs and other neuronal factors is different if not completely lacking in Human Embryonic Kidney (HEK) cells, the model of choice in this manuscript. These are a useful tool and can be used for preliminary screens, but they are definitely not neurons. The authors should keep this in mind when discussing their data, and even if I appreciate that repeating the whole BioID screen in neurons might not be feasible, at least validations in neurons with assays such as PLA are definitely needed.
- SMN, like many other RBPs is localised both in the nucleus and cytoplasm. The authors never show the localisation of their BirA-SMN1 construct and whether it has a similar localisation to endogenous SMN1. Similarly the authors should provide stainings looking at the localisation of the SMN mutants, as alterations in their localisation may impact on their interactome. Moreover, as the endogenous protein is localised both in the nucleus and cytoplasm, in addition to the NLS-BirA, the authors should have a NES-BirA control too. The SUMO-BirA may have a predominantly cytoplasmic localisation but the authors should at least show this.
- The authors investigate whether arginine methylation is important for SMN/FMRP binding. The authors find that silencing PRMT1 leads to a decreased interaction between SMN and FMRP. However, this could be a completely indirect effect, since the authors never test the methylation of the two proteins following PRMT1 knock down. Testing this or performing in vitro binding assay in the presence or absence of PRMT1 would clarify this point.
- In the last part of the manuscript the authors investigate the LLPS of SMN1. Hardly no information is given in the methods on how these assays are performed - with the only information given being: to induce droplet formation, SMN was concentrated on Amicon 3,000 MWCO column (Millipore) to ~90 μ M and stored at 4{degree sign}C. This makes it impossible to understand the nature of their assays and their results. Are the starting recombinant proteins (wt and mutants) soluble? Generally, IDR-rich proteins tend to aggregate and it is necessary to add cleavable tags to maintain them soluble prior to performing any in vitro LLPS assay. Is this the case with SMN?
- In Fig. 5C rather than liquid droplets SMN seems to form an aggregate. In fact FMRP seems to surround the structure, rather than partially partition within. In order to make any statement on the phase separation propensity of wt and mutant SMN the authors should consider performing turbidity assays and FRAP experiments.
- The authors do not provide any quantification (or n number) of any of the experiments presented, except the BioID. Quantification and number of independent experiments are necessary.

Other relevant points:

- To investigate SMN interactome the authors use BioID. This is a rather old system to perform proximity labelling and requires the addition of biotin for 15-20h. Newer versions, such as the widely used TurboID only requires 10 minutes and gives a labelling that is temporally much better controlled. The authors should specify why BioID is their system of choice.
- The authors state that "HEK293T cells, an immortal transformed human cell line, but which has interestingly neuron-like features, such as morphology and transcriptome". HEKs cells have a morphology that is very far from neurons and they definitely do not express neuronal specific proteins. They can be useful models, especially to perform preliminary screens.
- In all the IP/PD assays the authors only show the co-IPed proteins. Immunoprecipitated and pulled down proteins should also be shown in the western blots.
- The authors validate the SMN1 proxisome exclusively through IPs/PDs of overexpressed targets. Endogenous interaction between interacting proteins should also be tested either by co-IP or PLA.
- Expression/localization of wild type and mutant SMN constructs should be shown as a different localisation of the protein could result in alterations in the interactome.
- In order to validate the interaction between SMN and FMRP the authors perform co-IPs and pull downs. It is unclear why the authors use their bioID BirA-Flag-SMN construct for this purpose. Rather the authors should have purified the endogenous protein or at least an exogenously expressed construct lacking the ligase. Moreover, it is unclear why the authors have not tried to detect endogenous FMRP, which may have given a cleaner signal (in the control lanes).
- In general, for clarity, when a tagged protein is used the authors should describe it as X-tagged X protein i.e co-immunoprecipitation between FLAG-FMRP and myc-BirA-SMN.
- In Fig.3B is the ratio of PD (FMRP)/Input (SMN)? If so, this is not comparable and quantification should be within the PD samples GST or GST-SMN/FMRP.

- In Fig.4A siRNA PRMT2/3/4 should be confirmed by western blotting.
- "SMN TUDOR is essential for interactions with FMRP" to state this the authors should prove that in the absence of the domain the binding is lost, as other regions could also take part in the binding.

RESPONSE TO REVIEWER #1:

In the present manuscript, Binda and colleagues report on the establishment of a Bio-ID strategy to identify novel interaction partners of SMN in human cells. They list and further validate Fragile X mental (FMR) family member proteins as SMN interactors picked up after biotinylation in the vicinity of SMN. These interactions are shown to **depend on an intact TUDOR domain in SMN**. The authors further investigate a potential function of R-methylation on interaction between SMN and FMR family members using RNAi against a variety of R-methyl transferases. The knockdown of PRMT1 turns out to specifically stimulate interaction. Finally, the authors **provide evidence for phase separation of SMN using purified, recombinant SMN** as wt and SMA-associated mutant variants. SMN's phase separation seems to be promoted under conditions of co-condensation with RNA species, while mutants in the TUDOR domain negatively impact phase separation in vitro.

We thank very much the Reviewer for her/his valuable comments and suggestions. Here are our "point-by-point" response to the Reviewer's concerns:

Taken together, this is a mixed bag of observations that are, individually mostly well documented and described. The paper is **written well and easy to read**. However, I failed to find a consistent take home message from the data provided. What is the new insight into SMN function and/or into the molecular basis for SMA? I am listing major concerns, which, from my point of view, preclude publication of the manuscript at this point.

We appreciate the overall positive comment. As highlighted in the title, and emphasized in the revision, our study reveals that SMA-associated mutations of SMN protein obviate specific properties of SMN, namely its interaction with FMRP family members and its capacity to promote phase separation. We believe these observations are meaningful and worth further investigations to better understand the pathophysiology of SMA.

Regarding "new insight into SMN function and/or into the molecular basis for SMA" or "take home message"... the title of the article says it all. SMN forms droplets, never been shown before. SMA-linked mutations prevent SMN from forming droplets (and associating with FMR_{FM}).

1. The Bio-ID approach is well done in its biochemical and data analysis part. We see biotinylation of SMN complex proteins as "top hits". So far so good. However, a characterization of the cellular situation upon induction of the Bio-ID construct is missing, but would be essential to recapitulate the status of SMN in cells under conditions of biotinylation. What are relative levels of exogenous BirA-SMN to endogenous SMN? How does BirA-SMN localize (Cajal bodies?) and how does the "cloud" of biotinylated substrates develop over time when biotinylation starts?

We performed immunoblot experiments to compare the transfected protein expression with the endogenous protein. SMN is an abundant protein, it would be difficult to express more than there actually is even in HEK293T cells. As shown in new Figure S1A (see below), MYC-BirA-SMN is actually expressed at lower levels than endogenous SMN. Comparison was performed twice in independent experiments. Similarly, MYC-BirA-FMRP is not overexpressed compared to the endogenous form.

Localization is assumed to be fine since BirA-SMN biotinylates GEMIN2-8 and most classic partners, while BirA-SMN TUDOR mutants fail to biotinylate COIL.

2. The identification of FMR family proteins, as the authors state themselves in the discussion (see ref 43), as interaction partners of SMN is NOT novel although highlighted as proof of principle for the power of the Bio-ID approach to identify novel SMN interactors. This is at least irritating.

As the Reviewer rightly pointed out, SMN-FMRP interaction has been reported. However, we identify FMR_{FM} (FMRP, FXR1, and FXR2) as a family as new SMN partners. Furthermore, we put more highlight on the previous study by Piazzon et al., 2008 (referenced now in introduction section of the revised version). While Piazzon et al. nicely documented the importance of the C-terminal regions of FMRP and SMN, our study documents the importance of the TUDOR domain in FMRP-SMN interaction.

In addition, we extend the previous observation to other members of the FMR protein family, i.e. FXR1 and FXR2. We believe FMRP and the novel FMR_{FM} are worth exploring for their role in SMN functions.

3. The approach to further investigate the dependence of FMR-SMN interactions of "reader" domains (TUDOR in SMN) is a valid one and comes with a good start. However, it is not leading anywhere in the end. What do the interaction and regulation mean? The authors cite ref43 saying that that paper, albeit identifying the interaction of FMR with SMN, did not assign any functional experiments. The present study, unfortunately, does not either. I do appreciate the clear-cut dependence on the TUDOR domain, but this does not elucidate the function any further.

We appreciate the encouraging comment of the Reviewer. Although our data were missing a direct evidence of the functional interaction between SMN_{TUDOR} and FMRP, the encapsulation of FMRP_{IDR} within SMN/RNA droplets, together with the previously reported role of SMN-FMRP interaction in translation, both suggest a potential role of the phase condensation properties in translation regulation.

To investigate the potential role of SMF-FMRP interaction, we fractionated ribosomes from mouse MN1 cholinergic motor neuron cells. We observed a co-fractionation of Smn and Fmrp (Figure 4D and below). Interestingly, upon translation inhibition using puromycin, both Fmrp and Smn shifted to lighter fractions (Figure 4D), suggesting that Smn and Fmrp work together to regulate translationally active ribosomes. Interestingly, upon RNase A treatment, Fmrp shifted to lighter fractions than Smn (Figure 4D). Ribosomal proteins Rpl7 and Rps3 were used as controls.

[This figure has been removed by LSA editorial staff per authors' request]

4. The (interesting and clean) observation that PRMT1 knockdown specifically increases SMN-FMR interaction is not taken further but only comes up in the discussion with some rather vague speculations of what this may mean. Is it really the catalytical activity that plays a role here, or the protein itself, as the authors ask themselves but do not further test. Where do PRMT1 and SMN interact (nucleus?) to impact on communication with FMR proteins (cytoplasm?).

FMRP can be found in the nucleus, associated with chromatin and histone marks [PMID: 24813610]. SMN is found in Cajal bodies in the nucleus, but also in the cytoplasm. In the first version of our study, PRMT1 showed up in the proximity proteome, but we did not validate an actual interaction with SMN, nor with FMRP; it could be a transient interaction or a simple proximity.

To address the pertinent comment of the Reviewer, we assessed the association between purified recombinant HIS-tagged PRMT1 and various purified recombinant GST-tagged SMN truncations. Interestingly, the amino terminal region (SMN_{Nterm}) seemed to predominantly directly associate with PRMT1 while the TUDOR domain (SMN_{TUDOR}) seemed to also associate to some degree with PRMT1 (see Figure below). These results are however outside the scope of the manuscript are we decided to leave them out (at least for now).

5. The paper finally shifts gears and analyses phase separation of recombinant SMN. The stabilization of condensates via RNA is intriguing although the experiment is neither carefully quantified (single images without scale bars...) nor controlled, e.g. via an RNase treatment, using fluorescein alone or using SMN variants deficient in RNA binding.

To follow the advices thankfully proposed by the Reviewer, we performed additional experiments:

- Scale bars are now included in condensation assays in the presence or absence of RNA in the new Figures 5 and 6. All images were obtain by microscopy using a 40x objective. This is now specified in Figure Legend and in the Materials and Methods sections.

- Control experiments using fluorescein alone were performed, and the corresponding microscopic pictures were added (new **Figure S6C**). The fluorescein dye alone is diffused across the field (Figure below, panel B) compared to fluorescein-labelled RNA, which overlaps with SMN droplets (Figure below, panel A). Therefore, it was not easy to image. The images were taken from two experiments.

- RNAse A treatment does cleave RNA, but doesn't remove nucleotides or poly-nucleotides leftovers. We did try previously to pre-treat the RNA extracts with RNAse, but rSMN droplets were still stabilized over time. RNA-binding proteins, such as SRSF1 [<https://doi.org/10.1038/s41467-020-20481-w>] and hnRNPG [doi: 10.1093/nar/gku244], make contact with only 2-3 nucleotides on RNA molecules. We used no RNA as the control.

- Alternatively, we performed turbidity assays, as suggested by Reviewer 2 (please refer to our response to Comment #5 of Reviewer 2). The RNAse A treatment, as well as the absence of RNA, significantly reduced the optical density of rSMN-RNA solutions (new **Figure 5C**).

The Figure legends and the Methods' section have been modified accordingly.

Minor issues:

Figure 1A: how can a stronger interaction of Bir-NLS with coilin than SMN mutants be explained? In other words: are there (and why) different backgrounds in interaction in this experiment?

Figure 1A does not indicate interactions between any of the proteins investigated, only proximity. BirA_{NLS} levels are higher than BirA-SMN, which could explain why it biotinylates "better" COIL than BirA-SMN_{Y109C} or BirA-SMN_{E134K} mutants. Hypothetically, BirA_{NLS} biotinylates COIL "better" than BirA-SMN_{Y109C} or BirA-SMN_{E134K} mutants because it is concentrated in the nucleus, whereas SMN mutants are throughout the whole cell and do not associate with COIL.

Figure 1B: Using a TUDOR domain mutant to compare with wt SMN would have been very insightful...

We agree with the reviewer on this. For additional MS studies we used the SMN_{ST} mutant BirA-SMN_{Y109C} to compare with BirA-SMN (see below). For the comparison between BirA-SMN and BirA-SMN-Y109C, proteins considered differential were included in the Volcano plot only if significantly different between the 2 conditions but only if also significant compared to BirA alone control. Thus, a protein colored green in the Volcano plot only if enriched in BirA-SMN relative to BirA-SMN_{Y109C} and BirA-SMN relative to BirA alone.

Figure 2B: The SMN blot looks clean in inputs but not in the IP fractions, why is that?

SMN is an abundant protein, thus the clean signal in the input lanes. Abundance of SMN in the IP lanes is probably lower, thus needing longer exposure time, resulting in increased background signal.

Figure 4A and B: see 2B

Same answer as above.

RESPONSE TO REVIEWER #2:

In this paper entitled "SMA-linked SMN mutants prevent phase separation and SMN interactions with FMRP family members", Binda and colleagues have investigated by proximity labelling the interactome of the RNA binding protein (RBP) SMN1, a protein causative of the neuromuscular disease SMA. The authors have found that the RBPs FMRP, FXR1 and FXR2 interact with SMN1 and this binding is impaired in the absence of the methyltransferase PRMT1. Moreover, SMN forms RNA-containing liquid droplets in vitro, suggesting that SMN LLPS could be involved in the formation of cajal bodies within cells, and mutations in SMN lead to the formation of aggregate-like structures.

The topic investigated is of high interest, however the data, as presented in this manuscript, is very preliminary and the experiments presented do not always support the conclusions. A better characterisation of the system is needed, moreover, thorough validations and controls, as well as quantifications and n numbers are missing throughout the manuscript. For these reasons this manuscript should not be accepted in its current form.

We are grateful to Reviewer 2 for the constructive and comprehensive review of our submission, and for the recommendations proposed.

Main points:

1. Alterations in SMN1 cause alterations in motor neurons, ultimately leading to the neuromuscular disease SMA. Investigating how mutations in SMN1 alter its interactome is definitely of interest,

however the expression of disease-relevant RBPs and other neuronal factors is different if not completely lacking in Human Embryonic Kidney (HEK) cells, the model of choice in this manuscript. These are a useful tool and can be used for preliminary screens, but they are definitely not neurons. The authors should keep this in mind when discussing their data, and even if I appreciate that repeating the whole BioID screen in neurons might not be feasible, at least validations in neurons with assays such as PLA are definitely needed.

We agree with the Reviewer that HEK293 cells are not neurones. However, this cell line has been extensively used as a cell model for different issues and regulatory mechanisms related to neurones. We are here referencing peer-reviewed articles that suggest similarities in morphology, transcriptome, and possible neurological origin of the cell line. Shaw *et al.* remains highly cited in important journals:

- Cell Metabolism [doi.org/10.1016/j.cmet.2021.03.005]
- PNAS [doi.org/10.1073/pnas.2020124117]
- JBC [doi.org/10.1074/jbc.RA120.012618]
- Scientific Reports [doi.org/10.1038/s41598-019-56483-y]
- Molecular & Cellular Proteomics [doi.org/10.1074/mcp.TIR118.000800]
- Nature Communications [doi.org/10.1038/s41467-017-02720-9]

2. SMN, like many other RBPs is localised both in the nucleus and cytoplasm. The authors never show the localisation of their BirA-SMN1 construct and whether it has a similar localisation to endogenous SMN1. Similarly the authors should provide stainings looking at the localisation of the SMN mutants, as alterations in their localisation may impact on their interactome. Moreover, as the endogenous protein is localised both in the nucleus and cytoplasm, in addition to the NLS-BirA, the authors should have a NES-BirA control too. The SUMO-BirA may have a predominantly cytoplasmic localisation but the authors should at least show this.

This comment overlaps with Reviewer 1 comment #1 (please, see also our response to Comment #1 of Reviewer 1). We agree with Reviewer 2 that BirA-SMN and SMN mutants do not necessarily have a subcellular localization similar to that of wild-type SMN. We respectfully draw the Reviewer's attention to the fact that TUDOR mutations of SMN impair interactions between SMN and important partners such as COIL, FBL, and RNAPolIII.

As for the control, previously published studies have highly recommended to use BirA-NLS as control [doi.org/10.3389/fgene.2020.00450].

3. The authors investigate whether arginine methylation is important for SMN/FMRP binding. The authors find that silencing PRMT1 leads to a decreased interaction between SMN and FMRP. However, this could be a completely indirect effect, since the authors never test the methylation of the two proteins following PRMT1 knock down. Testing this or performing in vitro binding assay in the presence or absence of PRMT1 would clarify this point.

Our experiments show that silencing of PRMT1 does not lead to a decreased interaction between SMN and FMRP, but rather to an increase in interaction. As suggested by the Reviewer, we used purified recombinant FMRP_{IDR} in pulldown assays with GST-SMN and observed an interaction between the RGG region of FMRP (FMRP_{IDR}) and SMN TUDOR domain (SMN_{TUDOR}) (see also response to reviewer 1, point 4). Finally, the incorporation of FMRP_{IDR} within SMN droplets suggests that the proteins would interact directly, or via the total RNA present in the samples.

4. In the last part of the manuscript the authors investigate the LLPS of SMN1. Hardly no information is given in the methods on how these assays are performed - with the only information given being: to induce droplet formation, SMN was concentrated on Amicon 3,000 MWCO column (Millipore) to ~90 µM and stored at 4°C. This makes it impossible to understand the nature of their assays and their results. Are the starting recombinant proteins (wt and mutants) **soluble**? Generally, IDR-rich proteins

tend to aggregate and it is necessary to add cleavable tags to maintain them soluble prior to performing any in vitro LLPS assay. Is this the case with SMN?

As described in the methods section, recombinant SMN (SMN-MxE-CBP), was purified using a chitin-agarose matrix and the CBP tag cleaved through the intein region (MxE). Otherwise, the formation of SMN droplets was as simple as concentrating recombinant SMN and then incubating on ice, as depicted in **Figure S5A**. We did not observe aggregates with wild-type rSMN prior to concentration, after which rSMN formed droplets while rSMN_{Y109C} and rSMN_{Y130C} formed aggregates, as depicted in **Figure S6A**.

5. In Fig. 5C rather than liquid droplets SMN seems to form an aggregate. In fact FMRP seems to surround the structure, rather than partially partition within. In order to make any statement on the phase separation propensity of wt and mutant SMN the authors should consider performing **turbidity assays** and FRAP experiments.

Some images can be misleading because SMN droplets get often deformed by the coverslip on top of the slide during imaging, but are not aggregates. Aggregates can be seen in **Figure 6B-C** with SMN mutants as well as in **Figure S6A**.

As suggested by the Reviewer, we set up a turbidity assay with rSMN with or without total RNA in triplicates. Turbidity was assessed by optical density at 330 nm (OD₃₃₀) on a Synergy H1 plate reader over time (every minute for 15 minutes) in triplicate. Turbidity in No RNA control samples dropped rapidly (~0.2 OD₃₃₀ units within the first 3 minutes) and seemed to level off after the first 3 minutes,

while samples with 0.1 µg/µL total RNA took 15 minutes to lose ~0.2 OD₃₃₀ units (see Figure on the left). Total RNA was also treated with RNase A (6.7 µg/µL) prior to conducting turbidity assays. In the absence of RNA (No RNA control) rSMN absorbance decreased rapidly compared to samples containing RNA (0.1 µg/µL total RNA). Similarly, samples pre-treated with RNase A had lower turbidity index, which decreased like No RNA control samples. These results corroborate the microscopy observations (**Figure 5A**). The turbidity assays were performed 3 times (n=3), except for the RNase treatment (n=2) with different preparations of rSMN.

The following text was added:

"To further characterize SMN droplets, turbidity assays were performed. As observed under the microscope, SMN droplets are stabilized in the presence of total RNA compared to No RNA control samples (**Figure 5C**). Furthermore, RNase A pre-treatment of the RNA led to reduced turbidity (absorbance at OD 330 nm) similar to the No RNA control (**Figure 5C**), highlighting the importance of RNA in the maintenance of SMN droplets over time."

6. The authors do not provide any quantification (or n number) of any of the experiments presented, except the BioID. Quantification and number of independent experiments are necessary.

Immunoblotting signals are now quantified using ImageJ and now provided where relevant (i.e. **Figure 3 and 4**).

However, we believe that quantifying immunoblots requires a standard curve ran in parallel for each immunoblot. Besides, precise quantification might be misleading for the reader: adding twice the amount of proteins, for example, does not equate to a doubling in the quantification. See example below:

Other relevant points:

- To investigate SMN interactome the authors use BioID. This is a rather old system to perform proximity labelling and requires the addition of biotin for 15-20h. Newer versions, such as the widely used TurboID only requires 10 minutes and gives a labelling that is temporally much better controlled. The authors should specify why BioID is their system of choice.

There's also protein A-tagged TurboID that allow BioID on endogenous protein, but the technology became available after the experiments were done. We now clarify why we used standard BioID "The BioID proteomic approach is well-established, broadly used, and remains relevant (19, 20)".

- The authors state that "HEK293T cells, an immortal transformed human cell line, but which has interestingly neuron-like features, such as morphology and transcriptome". HEKs cells have a morphology that is very far from neurons and they definitely do not express neuronal specific proteins. They can be useful models, especially to perform preliminary screens.

Reviewer 1 has raised a comment on HEK293 cells (Main point #1). As we mentioned in our answer, HEK293 cells have been described with specific neuronal markers and characteristics. They have been extensively used as cell model in highly recognized publications dealing with neuronal regulatory mechanisms.

- In all the IP/PD assays the authors only show the co-IPed proteins. Immunoprecipitated and pulled down proteins should also be shown in the western blots.

All experiments with recombinant GST-tagged proteins show the pulled down GST levels. Not feasible to redo Figure 2A and 2B to show IP levels because FLAG-CSPR2 and FLAG-TIRR have a molecular weight of 25 kDa (size of light chain). We agree that ideally IP levels of IPed and co-IPed proteins should be shown, however it is not always feasible. Additionally, we show interaction (or proximity) between SMN and FMRP by (1) BioID, (2) pulldown of biotinylated FMRP by BirA-SMN over BirA alone, (3) co-IP of FLAG-FMRP and MYC-BirA-SMN, (4) GST pulldowns using recombinant SMN wild-type and mutant forms, and (5) a direct interaction between FMRP_{IDR} and GST-SMN_{TUDOR} (see additional **Figure 3B** in response to final point below), strongly supporting a genuine SMN-FMRP interaction. Figure 4A and 4B cannot be redone because the siRNA library is not available in Ottawa where the main author relocated. Pulldown levels of MYC-tagged BirA are not relevant as we were looking at biotinylated proteins being pulled down.

- The authors validate the SMN proxisome exclusively through IPs/PDs of overexpressed targets. Endogenous interaction between interacting proteins should also be tested either by co-IP or PLA.

We have re-done the co-IP for endogenous SMN and FMRP (previously not shown) to include IPed and input levels for both FMRP and SMN (to address previous point). Using HEK293T cell extracts we were able to observe an interaction between endogenous SMN and FMRP (new panel **Figure 4C**).

- Expression/localization of wild type and mutant SMN constructs should be shown as a different localisation of the protein could result in alterations in the interactome.

Reviewer 1 has raised a similar comment (please see our answer to Reviewer 1, Comment # 1). To address the Reviewer's concern, we now provide expression levels between endogenous SMN and BirA-tagged SMN (see **Figure S1A**).

- In order to validate the interaction between SMN and FMRP the authors perform co-IPs and pull downs. It is unclear why the authors use their bioID BirA-Flag-SMN construct for this purpose. Rather the authors should have purified the endogenous protein or at least an exogenously expressed construct lacking the ligase. Moreover, it is unclear why the authors have not tried to detect endogenous FMRP, which may have given a cleaner signal (in the control lanes).

We reasoned that BirA alone could be used as actual negative control (instead of a nearly empty vector), just like GST alone is used as a negative control in GST pulldown experiments.

We now have performed coIP between endogenous SMN and FMRP (see previous point and **Figure 4C**).

- In general, for clarity, when a tagged protein is used the authors should describe it as X-tagged X protein i.e co-immunoprecipitation between FLAG-FMRP and myc-BirA-SMN.

We agree with the Reviewer's comment. We specified in the revision FLAG- and MYC- tags wherever they were missing (highlighted in blue in the revised version).

- In Fig.3B is the ratio of PD (FMRP)/Input (SMN)? If so, this is not comparable and quantification should be within the PD samples GST or GST-SMN/FMRP.

We apologize for the lack of clarity. In fact, the ratio is between pulled down GST-SMN and FMRP signal. This is now specified in the Figure legend "Ratios below panel 3B are between pulled down GST-SMN and pulled down FLAG-FMRP".

- In Fig.4A siRNA PRMT2/3/4 should be confirmed by western blotting.

The siRNA used were published by Sabra *et al* in 2013 and validated by immunoblotting. Current antibodies for PRMT2-4 did not work in our hands unfortunately.

- "SMN TUDOR is essential for interactions with FMRP" to state this the authors should prove that in the absence of the domain the binding is lost, as other regions could also take part in the binding.

TUDOR mutants broadly impaired SMN-FMRP and SMN-FXR1 (**Figure S3**), thus an intact TUDOR domain is essential for interactions with FMRP. To address this issue, we expressed SMN truncations (GST-SMN_{N-term}, GST-SMN_{TUDOR}, and GST-SMN_{C-term} [**Figure 3A**]) to assess the requirement of the TUDOR domain as a domain.

The following text was added: " To address which region of SMN associates with FMRP, GST-tagged SMN truncations were generated (represented in **Figure 3A**). Only the RGG-containing region of FMRP (amino acid residues 445-590 [FMRP_{IDR}]) could be expressed and purified. As expected, recombinant SMN_{TUDOR} could on its own associate directly with purified T7-tagged FMRP_{IDR} (**Figure 3B**).

September 28, 2022

Re: Life Science Alliance manuscript #LSA-2022-01429-TR

Dr. Olivier Binda
University of Ottawa
Department of Cellular and Molecular Medicine
451 Smyth Road
Ottawa, Ontario K1H 8M5
Canada

Dear Dr. Binda,

Thank you for submitting your revised manuscript entitled "SMA-linked SMN mutants prevent phase separation and SMN interactions with FMRP family members" to Life Science Alliance. The manuscript has been seen by one of the original reviewers whose comments are appended below. While the reviewer continues to be overall positive about the work in terms of its suitability for Life Science Alliance, one important issue remains.

In the first round of review, both Reviewers asked to show the localization of BirA-SMN and whether it has a similar localization to endogenous SMN1. Considering that so much of the study is based on the expression of this protein, I think this is important to show.

Our general policy is that papers are considered through only one revision cycle; however, given that the suggested changes are relatively minor, we are open to one additional short round of revision. Please note that I will expect to make a final decision without additional reviewer input upon re-submission.

Please submit the final revision within one month.

To upload the revised version of your manuscript, please log in to your account: <https://lsa.msubmit.net/cgi-bin/main.plex>
You will be guided to complete the submission of your revised manuscript and to fill in all necessary information.

-- A letter addressing this final point.

B. MANUSCRIPT ORGANIZATION AND FORMATTING:

Sincerely,

Reviewer #1 (Comments to the Authors (Required)):

Binda and colleagues have undertaken a rapid, comprehensive and successful revision of their manuscript on "SMA-linked SMN mutants prevent phase separation and SMN interactions with FMRP family members". I appreciate several issues that have been experimentally addressed or clarified in the text. In particular, data on ribosome fractionation of SMN and FMRP in Fig. 4D, the data provided for reviewers on the interaction between SMN and PRMT1 and SMN-RNA co-condensation shown by turbidity assays in Fig. 5C. Taken together, all issues have been fully addressed and I am happy to recommend publication of the manuscript in its present form.

October 17, 2022

RE: Life Science Alliance Manuscript #LSA-2022-01429-TRR

Dr. Olivier Binda
University of Ottawa
Department of Cellular and Molecular Medicine
451 Smyth Road
Ottawa, Ontario K1H 8M5
Canada

Dear Dr. Binda,

Thank you for submitting your revised manuscript entitled "SMA-linked SMN mutants prevent phase separation and SMN interactions with FMRP family members". We would be happy to publish your paper in Life Science Alliance pending final revisions necessary to meet our formatting guidelines.

- please upload your supplementary figure files as single files
- please add ORCID ID for first corresponding author-you should have received instructions on how to do so
- please add the Twitter handle of your host institute/organization as well as your own or/and one of the authors in our system
- please add a legend for your video to the main manuscript text
- please use the [10 author names, et al.] format in your references (i.e. limit the author names to the first 10)
- please double-check your callouts for Figure S5; you have a callout for panels D-F, but these panels are not in the legend or the figure
- PRIDE identifier PXD030970 does not pull up a submission, is this in process?

Figure Check:

- Figure 5A, Figure 6A, B, C, Figure S5 B,C, Figure S6A need scale bars

A. FINAL FILES:

B. MANUSCRIPT ORGANIZATION AND FORMATTING:

Sincerely,

-please upload your supplementary figure files as single files

The video is in Figure S5, but since supps have to be PDF, not PPT files, the video doesn't play and had to be uploaded separately.

I'm not sure I understand this point. Should all supplementary data merged into 1 single file?

-please add ORCID ID for first corresponding author-you should have received instructions on how to do so.

My ORCID ID 0000-0002-1539-0828 is now linked with the submission portal.

-please add the Twitter handle of your host institute/organization as well as your own or/and one of the authors in our system.

@uOttawa

@uOttawaMed

@UnivLyon1

@iNeuroMyoGene

-please add a legend for your video to the main manuscript text

The legend is in Figure S5C:

"(C) rSMN was labelled with Cy3 for visualisation of internal structures in the presence of RNA (0.1 $\mu\text{g}/\mu\text{L}$). Pictures taken 60 seconds apart were assembled into a short video."

Because the PowerPoint file containing the supplementary figures, including the video S5C, had to be converted to PDF, the video became still, so I submitted the video separately...

I added the following precision in the legend:

(see supplementary S5C animation in separate .AVI file).

-please use the [10 author names, et al.] format in your references (i.e. limit the author names to the first 10).

Downloaded LSA EndNote style and applied.

-please double-check your callouts for Figure S5; you have a callout for panels D-F, but these panels are not in the legend or the figure.

Corrected. Replaced S5D-F callout to Figures 6B-C and S6A.

-PRIDE identifier PXD030970 does not pull up a submission, is this in process?

The data was deposited, but not made publically available before publication. It is now publically accessible.

Figure Check:

-Figure 5A, Figure 6A, B, C, Figure S5 B,C, Figure S6A need scale bars.

Scale bars have been added to Figures 5A, 6A-C, S5B-C, and S6A.

No plans for press release.

No plans for video.

A. FINAL FILES:

Manuscript is in .docx format.

All main figures are in .ai Illustrator CS6 format.

Our summary blurb is 197 character long.

October 25, 2022

RE: Life Science Alliance Manuscript #LSA-2022-01429-TRRR

Dr. Olivier Binda
University of Ottawa
Department of Cellular and Molecular Medicine
451 Smyth Road
Ottawa, Ontario K1H 8M5
Canada

Dear Dr. Binda,

Thank you for submitting your Research Article entitled "SMA-linked SMN mutants prevent phase separation and SMN interactions with FMRP family members". It is a pleasure to let you know that your manuscript is now accepted for publication in Life Science Alliance. Congratulations on this interesting work.

DISTRIBUTION OF MATERIALS:

Again, congratulations on a very nice paper. I hope you found the review process to be constructive and are pleased with how the manuscript was handled editorially. We look forward to future exciting submissions from your lab.

Sincerely,
